# Introducing pre-exposure prophylaxis to prevent HIV acquisition among men who have sex with men in Sweden: insights from a mathematical pair formation model

Disa Hansson ![ORCID],[1] Susanne Strömdahl,[2,3] Ka Yin Leung,[1] Tom Britton[1]

[1]Department of Mathematics, Stockholm University, Stockholm, Sweden
[2]Department of Medical Sciences, Uppsala University, Uppsala, Sweden
[3]Department of Public Health Sciences, Karolinska Institute, Stockholm, Sweden

**Correspondence to**
Dr Disa Hansson; disa@math.su.se

## ABSTRACT

**Objectives** Since 2017, the Public Health Agency of Sweden recommends that pre-exposure prophylaxis (PrEP) for HIV should be offered to high-risk individuals, in particular to men who have sex with men (MSM). The objective of this study is to develop a mathematical model investigating the effect of introducing PrEP to MSM in Sweden.

**Design** A pair formation model, including steady and casual sex partners, is developed to study the impact of introducing PrEP. Two groups are included in the model: sexually high active MSM and sexually low active MSM. Three mixing assumptions between the groups are considered.

**Setting** A gay-friendly MSM HIV/sexually transmitted infection testing clinic in Stockholm, Sweden. This clinic started offering PrEP to MSM in October 2018.

**Participants** The model is calibrated according to detailed sexual behaviour data gathered in 2015 among 403 MSM.

**Results** By targeting sexually high active MSM, a PrEP coverage of 3.5% of the MSM population (10% of all high actives) would result in the long-term HIV prevalence to drop considerably (close to 0%). While targeting only low actives would require a PrEP coverage of 35% for a similar reduction. The main effect of PrEP is the reduced susceptibility, whereas the increased HIV testing rate (every third month) among PrEP users plays a lesser role.

**Conclusions** To create a multifaceted picture of the effects of interventions against HIV, we need models that include the different stages of HIV infection and real-world data on detailed sexual behaviour to calibrate the mathematical models. Our findings conclude that targeting HIV high-risk individuals, within HIV risk populations such as MSM, with PrEP programmes could greatly decrease the long-term HIV prevalence in Sweden. Therefore, risk stratification of individuals is of importance in PrEP implementation programmes, to ensure optimising the effect and cost-effectiveness of such programmes.

## INTRODUCTION

In Sweden, the HIV prevalence was estimated to be 0.07% in the general population in 2015 (<8000 individuals),[1] whereas the

### Strengths and limitations of this study

► Using a mathematical pair formation model we study the effect of introducing pre-exposure prophylaxis (PrEP) among men who have sex with men (MSM) in Sweden, a group at high risk of HIV acquisition.

► The model divides the population into sexually high active MSM and low active MSM, where high actives are offered to use PrEP.

► The model is calibrated to detailed sexual behavioural data gathered among the MSM population now being offered PrEP in Sweden.

► Limitations of this study include that the data only make it possible to include two activity groups, and that we do not allow for more than one steady sex partner at a time.

self-reported HIV prevalence among men who have sex with men (MSM) has been estimated between 2% and 6%.[2] Sweden was the first country to report having achieved the Joint United Nations Programme on HIV/AIDS/WHO 90-90-90 goal in 2016,[1] with at least 90% of people living with HIV being aware of their HIV status, 90% of HIV-diagnosed individuals being on antiretroviral therapy (ART) and with 90% of those on ART being under viral suppression.[3] Viral suppression means achieving continuously undetectable HIV viral load that diminishes onward transmission to close to zero.[4 5]

As a result of ART's effectiveness in viral suppression it can be viewed as an effective preventive measure for further HIV transmission. However, on its own it does not seem to reduce HIV prevalence enough in risk groups, such as the MSM population, but needs to be combined with additional preventive strategies.[6–8] One such preventive intervention is oral pre-exposure prophylaxis (PrEP) for HIV, that is, that the

antiviral drug tenofovir-emtricitabine is taken by individuals with negative HIV serostatus to prevent HIV acquisition.[9 10] PrEP effectiveness is dependent on adherence to PrEP to ensure that protective concentrations of the drugs are present at exposure to prevent transmission.[11] Two different studies report that PrEP reduces the HIV incidence by 86% among MSM (95% CI 40 to 98% and 90% CI 64 to 96, respectively).[9 10] Due to the effectiveness of PrEP, the WHO recommends PrEP to be offered to individuals at substantial risk of HIV acquisition, defined as an HIV incidence of 3 or above per 100 person-years.[12]

The use of PrEP in Sweden was approved in 2016 by the Swedish Medical Products Agency.[13] Since 2017, the Public Health Agency of Sweden recommends that PrEP should be offered to high-risk MSM. However, very few clinics started offering PrEP at this time due to logistical and funding concerns. Since June 2018, the New Therapies Council recommends Swedish counties to implement PrEP programmes for MSM and offer subsidised PrEP.[14] The larger gay-friendly sexual health clinics in the major three urban areas of Sweden have since then started to implement PrEP. As of July 2019, approximately 315 individuals have initiated PrEP in Stockholm (Dr Finn-Filén responsible for PrEP at Venhälsan, Södersjukhuset, personal communication).

The objective of this study is to investigate the effect of introducing PrEP to MSM in Sweden. First, we incorporate the use of PrEP into pair formation models developed to study HIV transmission.[15] This model separates individuals depending on sexual activity degree, high active or low active. The model is then fitted to sexual behaviour data from a gay-friendly HIV/sexually transmitted infection (STI) testing clinic in Stockholm, Sweden. Finally, the effect of PrEP on HIV transmission is studied by risk-stratifying MSM for PrEP, to explore the level of PrEP coverage needed to substantially reduce the long-term HIV prevalence.

## METHODS

To study the introduction of PrEP, we develop a pair formation model that includes steady (long-term) partnerships and casual (one-off/occasional) sex partners. We categorise individuals as sexually high active or low active, with different sex partner mixing patterns, different HIV diagnosis rates and allowing for high actives to use PrEP. We incorporate two stages of HIV infectiousness: the early acute (primary) stage and the subsequent chronic (asymptomatic) stage.[16] The model is then calibrated to detailed sexual behaviour data and the observed HIV prevalence. We earlier developed a model to disentangle the roles of casual and steady partnerships on HIV transmission, which we further elaborate in this work.[15] We begin by describing the model for sexual contacts and continue with how transmission of HIV infection is modelled. Then we apply sexual behaviour data gathered among MSM. The full description of the model can be found in the online supplementary material S1.

### Dynamic pair formation model

Consider a sexually active same-sex population where new individuals enter the sexually active population without having a steady sex partner. Individuals can have at most one steady partner at a time, which can end by separation or death of either partner. Individuals can also have casual sex partners during steady partnerships as well as during periods without steady partnerships; the rate at which this occurs depends on the steady partnership status of the individuals under consideration. Based on data, we allow singles to have a higher rate of casual sex than individuals in a steady partnership. Not only partnership status affects the rate of finding new casual sex partners, we additionally allow for individuals to be either high active or low active regarding the frequency of having casual sex partners.

Letting the rate of finding a new casual sex partner depend upon the partnership status and the activity degree of both potential members in the sexual act, yields 16 different casual sex partnership combinations. We let $\alpha_{ij}^{rq}$ denote the rate at which an individual with activity degree $r$ and $i$ steady partners try to find a casual sex partner with activity degree $q$ and $j$ steady partners, where $r, q \in \{h = \text{high}, l = \text{low}\}$ and $i, j \in \{0, 1\}$. These rates play an important role in our modelling and are described in detail in online supplementary material S1 and S2.

### Casual sex and mixing patterns

Creating the groups high active and low active makes it necessary to formulate mixing between the groups. Three activity degree mixing assumptions are considered: proportionate mixing, complete assortativity and mixing fitted to responses of a proxy question. Common to all three models is that high actives have casual contacts at a fixed rate being larger than that of low actives. Proportionate mixing implies that an individual chooses a casual partner at random among the potential casual sex attempts in the population, that is, the probability of having a high active casual partner is the same for low active and high active individuals. Complete assortativity means that high actives only have high active casual partners and low actives only have low active casual partners.

We estimate the assortativity according to the testing clinic participants' answer to one question on partners' sexual activity, as described in online supplementary material S3. This question is referred to as a proxy question for partners' activity degree. The proxy for partners' activity degree and knowing whether an individual is high active or low active specifies the amount of assortativity in the data. The estimated assortativity can take values between 0 and 1, where 0 corresponds to proportionate mixing and 1 to complete assortativity.

Since we do not know whether or not participants' casual sex partners are in steady partnerships, the mixing between singles and individuals in a partnership is assumed to be random (proportionate mixing).

## Model of infection

To model the spread of infection we use a compartmental model, much like the Susceptible → Infectious → Recovered model,[17] but with two stages of infectiousness: acute followed by chronic. The probability of transmission of HIV depends on the stage of infection when no antiretroviral treatment is used.[18] In the early acute stage the probability of transmission is much higher due to higher viral load than in the chronic stage. The probability of transmission in one unprotected sex act (in our case anal intercourse) is denoted $p_A$ when in the acute stage and $p_C$ when in the chronic stage. The model allows for different levels of condom use with steady and casual sex partners, and it is incorporated by reducing the transmission probability accordingly. The time until diagnosis and the beginning of ART depends on the degree of sexual activity. Further, individuals on ART are assumed to be virally suppressed and thereby to no longer transmit infection. The compartmental model therefore divides the population into susceptible, infectious in the acute stage, infectious in the chronic stage and being on ART treatment.

The aim with defining this model is to introduce the possibility for high actives to take PrEP, which, when taken correctly, decreases the probability of getting infected with HIV by approximately 86%.[9 10] Moreover, individuals accepting PrEP need to test themselves every third month. The model is illustrated in online supplementary material S1.

## Data and calibration

The data were gathered at a gay-friendly HIV/STI testing clinic (Venhälsan) in Stockholm, Sweden, during 2015.[19] MSM visiting testing clinics might be more sexually active than other MSM, for example, sexually inactive or MSM with one sex partner might not visit testing clinics as often. Four hundred and three MSM participants answered a structured timeline follow-back questionnaire and reported their total number of sex partners during the last 12 months. Detailed sexual behaviour data were collected on participants' last 10 sex partners during the last 12 months, including: type of sex partner (casual or steady); frequency of sex acts; condom use with each partner; the duration of each sexual relationship; and the answer to the proxy question on partners' sexual activity degree. All 403 participants were included in the study by Hansson et al.[15] However, inclusion in this study requires that participants have reported their total number of sex partners during a year to determine their activity degree. Moreover, for a partner to be included, the proxy question regarding the partner's sexual activity degree must be answered. Of the 403 participants, four reported having zero sex partners the previous 12 months and 28 participants did not answer that particular question. Of the remaining 371 participants, detailed information on 1991 different sex partners (510 steady and 1481 casual) was given. We have an answer to the proxy question for 1424 of the 1481 casual sex partners. When removing the

57 casual sex partners with no answer to the proxy question, 368 participants (and 1903 partners) remain and were included in the analysis presented here.

The participants demonstrate a considerable difference in yearly number of sexual partners, with a range from 1 to 250 sexual partners. We choose the mean (15) as our cut-off for defining high active individuals, resulting in that 33.7% are high actives. The mean number of sex partners for high actives is 33.2 (SD=32) and for low actives it is 6.0 (SD=3.2).

### Partnership and epidemic parameters

The parameters and their values used in the analysis are given in table 1. Some parameters are estimated from the testing clinic data (parts 1–4 in table 1), some parameters are varied (part 2 in table 1) and some are taken from the available literature (part 5 in table 1). In the analysis, the SEs of all estimates from the testing clinic data can be included to obtain a 95% credibility interval (CRI; using Monte Carlo simulation) of the prevalence estimated from the model (see online supplementary material S3).

The mean times to ART, denoted $\gamma_h$ for a high active and $\gamma_l$ for a low active, are calibrated to fit the observed prevalence. From the data we estimate that $\gamma_h = 2.35\gamma_l$ (see online supplementary material S3), this relationship will be kept throughout the analysis, such that only one of the parameters needs to vary.

### Estimated rates of meeting a new casual sex partner

Table 1 shows the estimated rates of meeting a new casual sex partner. In the third part of table 1 we have not used the proxy question on partners' activity degree, these estimates are enough when assuming proportionate mixing or complete assortativity. The proxy question is used to get the estimates in the fourth part of table 1. From this table and a given choice of mixing pattern, one can estimate the rates of looking for a new casual sex partner $\alpha_{ij}^{rq}$. The final values of $\alpha_{ij}^{rq}$ can be found in online supplementary material S3.

### Patient and public involvement

There was no patient and/or public involvement in the planning of this study.

## ANALYSIS AND RESULTS

The assortativity regarding activity degree is measured to 0.14, meaning that the studied population choose casual partners with a moderate assortativity. The HIV prevalence among the 368 participants is 5%, which is in line with national levels among MSM,[2] and this prevalence will be used as a baseline when studying the effect of PrEP. Specifically, the baseline model is the model where no one uses PrEP ($\xi = 0$) and that is calibrated to achieve a 5% equilibrium prevalence. The only parameter not being able to be estimated from the HIV/STI testing clinic data, or that can be taken from the literature, is the mean time to successful ART treatment. Hence, to calibrate the

**Table 1** Estimates of partnership and epidemic parameters

| Parameter | Value | Definition | Source |
|---|---|---|---|
| **1. Partnership parameters** | | | |
| $1/(\sigma + 2\mu)$ | 271.5 days | Mean duration steady partnership | MSM data |
| $1/\rho P_0$ | 152.8 days | Mean time being single | MSM data |
| $1/\lambda$ | 12.3 days | Mean time between AI within steady partnership | MSM data |
| $P_0$ | 0.360 | Fraction without a steady partner | MSM data |
| $P_1$ | 0.640 | Fraction with a steady partner | MSM data |
| $\pi_h$ | 0.337 | Fraction high actives | MSM data |
| $\pi_l$ | 0.663 | Fraction low actives | MSM data |
| **2. Parameters for condom use, PrEP and time to treatment** | | | **Source** |
| $q_s$ | 54.1% | Mean condom use steady partner | MSM data |
| $q_c$ | 62.9% | Mean condom use casual partner | MSM data |
| $\xi$ | | Rate for a high active to start taking PrEP (calibrated to achieve different % PrEP coverage) | Varied |
| | | *Mean time from infection to successful antiretroviral treatment for a:* | |
| $1/\gamma_P$ | 0.25 years | High active on PrEP | 27 |
| $1/\gamma_h$ | 1.5–3 years | High active not on PrEP | Varied |
| $1/\gamma_l$ | 3.5–7 years | Low active | $1/\gamma_l = 2.35/\gamma_h$ |
| **3. Casual sex partner parameters from data not using proxy** | | | **Source** |
| | | *Mean time until new casual sex partner for a:* | |
| $1/\alpha_{0.}^{h:}$ | 10.7 days | High active when single | MSM data |
| $1/\alpha_{1.}^{h:}$ | 12.5 days | High active when in partnership | MSM data |
| $1/\alpha_{0.}^{l:}$ | 66.5 days | Low active when single | MSM data |
| $1/\alpha_{1.}^{h:}$ | 97.9 days | Low active when in partnership | MSM data |
| **4. Casual sex partner parameters from data using proxy** | | | **Source** |
| | | *Mean time until new casual partner for a high active with a:* | |
| $1/\alpha_{0.}^{hh}$ | 14.1 days | High active when single | MSM data |
| $1/\alpha_{0.}^{hl}$ | 43.7 days | Low active when single | MSM data |
| $1/\alpha_{1.}^{hh}$ | 15.4 days | High active when in partnership | MSM data |
| $1/\alpha_{1.}^{hl}$ | 66.4 days | Low active when in partnership | MSM data |
| | | *Mean time until new casual partner for a low active with a:* | |
| $1/\alpha_{0.}^{lh}$ | 109.3 days | High active when single | MSM data |
| $1/\alpha_{0.}^{l}$ | 169.6 days | Low active when single | MSM data |
| $1/\alpha_{1.}^{lh}$ | 136.1 days | High active when in partnership | MSM data |
| $1/\alpha_{1.}^{l}$ | 348.9 days | Low active when in partnership | MSM data |
| **5. Parameters from the literature** | | | |
| | 70% | Condom efficiency | 28 |
| $1/\delta_a$ | 0.24 years | Mean time in acute infection stage | 18 |
| $1/\mu$ | 60 years | Sexually active lifespan | 29 30 |
| | | *Per-act transmission probability* | |
| | 0.1835 | Acute stage URAI | 16 |

Continued

**Table 1** Continued

| Parameter | Value | Definition | Source |
|---|---|---|---|
| | 0.0138 | Chronic stage URAI | 16 |
| | 1.48% | Overall URAI | 31 |
| | 0.62% | Overall UIAI | 31 |
| | 2.39 | Of URAI in comparison to UIAI | 31 |
| $p_A$ | 0.1301 | Acute stage combined URAI-UIAI | 16 31 |
| $p_C$ | 0.0098 | Chronic stage combined URAI-UIAI | 16 31 |

AI, anal intercourse; MSM, men who have sex with men; PrEP, pre-exposure prophylaxis; UIAI, unprotected insertive anal intercourse; URAI, unprotected receptive anal intercourse.

model to the observed 5% prevalence, we find the mean time to treatment that corresponds to this prevalence. To study the effect of PrEP, we use the same parameter set-ups as for the baseline model but additionally allow for sexually high actives to use PrEP ($\xi > 0$), and then find the new equilibrium prevalence. We use 'long-term prevalence' and 'equilibrium prevalence' interchangeably. The set of equations used to find the equilibrium prevalence can be found in online supplementary material S4. We begin by presenting the results of the model where PrEP has not yet been introduced, then we move to the model where high actives are offered PrEP.

### Prior to the introduction of PrEP

In dividing the population into two activity degrees and using fitted assortativity, we find that the prevalence of 5% (95% CRI 2.3% to 7.6%) is obtained when the mean time to ART is $\gamma_h^{-1} = 1.77$ years for high actives (4.15 years for low actives). In doing this calibration to the observed 5% prevalence, the estimated per cent of individuals with positive HIV serostatus that are on ART treatment is 95.8%.

Disregarding the proxy question, we would not know how the population mix regarding activity degree. We could then use the other two mixing patterns. Assuming proportionate mixing, the prevalence of 5% (95% CRI 2.1% to 7.8%) is obtained when $\gamma_h^{-1} = 1.79$ years. This set-up yields that the estimated per cent of individuals with HIV that are on ART treatment is 95.7%. For complete assortativity, the prevalence of 5% (95% CRI 2.7% to 7.0%) is obtained when $\gamma_h^{-1} = 1.63$ years, and then the estimated per cent of individuals with HIV that are on ART treatment is 96.4%. Figure 1A depicts the prevalence for varied values of the mean time to ART treatment and figure 1B shows the 95% CRIs.

Using the fitted assortativity, a prevalence of 5% was found when the mean time to ART was 1.77 years for high actives, while the same time to ART for the proportionate mixing assumption yields a prevalence of 4.6% (95% CRI 1.7% to 7.3%), and complete assortativity yields a prevalence of 6.9% (95% CRI 4.6% to 8.8%). This shows that higher assortativity regarding activity degree leads to higher prevalence and easier allows for HIV being endemic. With increased assortativity, the allocation of

the infected individuals becomes different, as seen in table 2, with more high actives being infected. Interesting to note, from figure 1A, is that the difference between the mixing assumptions decreases with an increased time to ART.

Additionally, from table 2 we note that approximately 35% of HIV transmissions occur within a steady partnership.

### Effect of introducing PrEP

We now present the effect of introducing PrEP. This is done by starting at a prevalence of 5% and then increasing the PrEP coverage. That is, we use the parameter values from the model without PrEP that achieved a 5% equilibrium prevalence, but now allow high actives to take PrEP and find the new equilibrium prevalence. We stress that the results are the long-term effect of certain levels of PrEP coverages, even with no more infections it will naturally take a long time to eliminate HIV completely from the MSM community, that is, for the HIV prevalence to reach 0%.

If the population is not divided according to activity degree, the PrEP coverage would need to be 5.2% of the total population to reduce the long-term prevalence from the observed 5% to close to 0% (see online supplementary material S5). Dividing the population according to activity degree and only targeting high actives for PrEP, figure 2A shows the combined effect of PrEP and an increased testing rate: reaching a coverage of 1% of the population (3.0% of high actives) will reduce the long-term prevalence from 5% to 3.6%; reaching a coverage of 3.5% of the population (10.4% of high actives) will reduce the long-term prevalence to 0%.

Being able to target risk groups for PrEP makes a big difference: targeting low actives instead would result in a needed PrEP coverage of 34.4% of the population to eventually eliminate HIV from the community (see online supplementary material S5).

To ascertain the respective effects of PrEP, the decreased susceptibility by 86% and the more frequent HIV testing rate (every third month), we do two additional analyses. If being on PrEP is not combined with an increased testing rate, but only a reduced susceptibility, reaching a coverage of 3.5% of the population will reduce the prevalence to

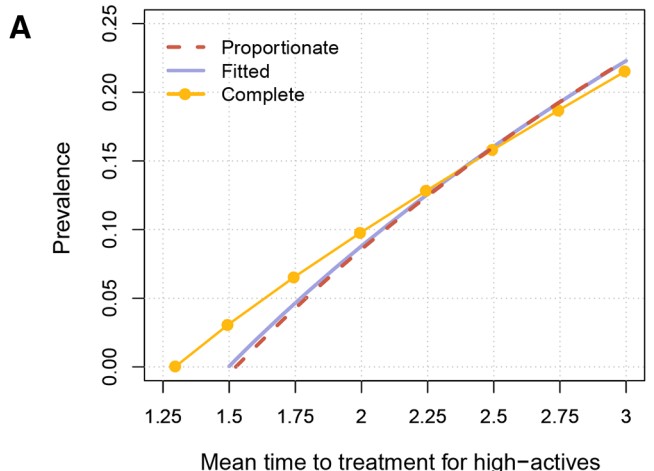

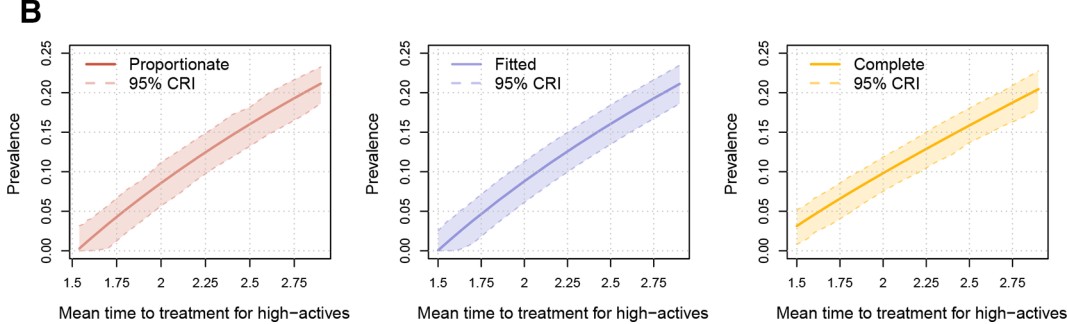

**Figure 1** (A) Estimated prevalence of HIV (y-axis) for the three mixing assumptions and different mean times to antiretroviral therapy (ART) treatment (x-axis). The presented time to ART treatment is for high active individuals, for low actives it is a factor 2.35 higher. (B) The same estimated prevalence as in (A), but now showing the prevalence separately and including the 95% credibility interval (CRI) for the three mixing assumptions. In one simulation, each partnership parameter (estimated from data) was drawn from its distribution. With that set-up of drawn parameters, we calculated the prevalence. This was repeated 1000 times to obtain the credibility interval.

**Table 2** For the three mixing assumptions, we show the estimated mean time to ART corresponding to a prevalence of 5%. For this prevalence and for each of the three mixing assumptions, we also show the route of transmissions and HIV prevalence for the two respective activity degree groups. The shown values for the time to ART treatment are for high active individuals, the time to ART treatment for low actives is 2.35 times larger. For the allocation of the 5% infected we show the percentage of high actives and low actives, respectively, that are HIV positive

|  | Overall HIV prevalence of 5% | | |
|---|---|---|---|
|  | **Proportionate mixing** | **Fitted assortativity** | **Complete assortativity** |
| Time to ART (years) | 1.79 | 1.77 | 1.63 |
| Route of transmission |  |  |  |
| Steady partner | 35% | 35% | 32% |
| Casual sex when in partnership | 38% | 39% | 41% |
| Casual sex when single | 26% | 26% | 27% |
| HIV prevalence in the group |  |  |  |
| High actives | 9.05% | 9.23% | 10.79% |
| Low actives | 2.94% | 2.85% | 2.06% |

ART, antiretroviral therapy.

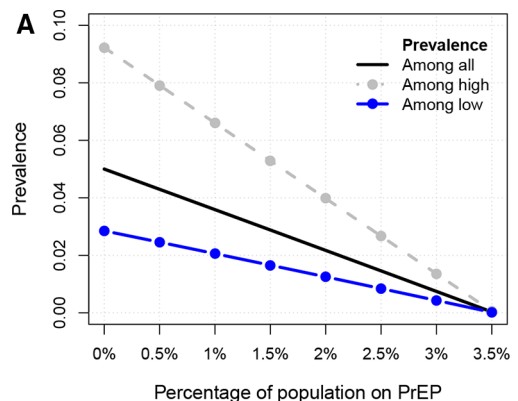
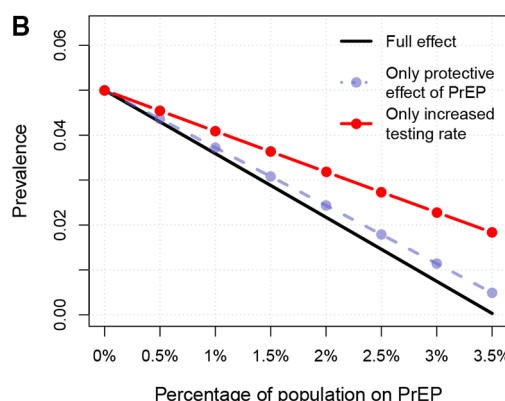

**Figure 2** The effect of introducing pre-exposure prophylaxis (PrEP) to sexually high actives. In (A) the x-axis shows different PrEP coverage levels and the y-axis the corresponding HIV prevalence. The three lines show: the HIV prevalence in the total population (black solid line), the HIV prevalence among high actives (lighter short-dashed line) and the HIV prevalence among low actives (darker long-dashed line). (B) Depicts the effect of PrEP by looking at: (1) solely the reduction of susceptibility and no increased testing rate; and (2) solely the increased testing rate and no reduced susceptibility.

0.5% (figure 2B). If being on PrEP does not give any reduced susceptibility, but only an increased testing rate, reaching a coverage of 3.5% of the population will reduce the prevalence to 1.9%.

For results concerning the more short-term effect of PrEP, see online supplementary material S5. We can, for example, note that using the lowest PrEP initiation rate that corresponds to HIV elimination (the 3.5% PrEP coverage) would after 50 years have reduced the prevalence from 5% to 4% and the percentage infectious and undiagnosed HIV cases by half (from 0.21% to 0.09%). Using a higher PrEP initiation rate would speed up the decrease in new HIV cases, for example, by reaching a PrEP coverage of 7% after 50 years would result in that the percentage of infectious undiagnosed HIV cases would be as low as 0.01%.

## DISCUSSION

Our results suggest that a PrEP coverage of at least 3.5% of the MSM population, when sexually high actives are targeted, is needed to eventually eliminate HIV from the MSM population in Sweden. This can be compared with a 34.4% PrEP coverage needed if only low active MSM were targeted for PrEP. These results emphasise the need for risk stratification among MSM, to ensure that those in need of PrEP receive the intervention. To reach high-risk MSM, outreach programmes and peer education programmes have been found to be effective, scale-up of these may increase the effect of PrEP implementation on the HIV epidemic among MSM in Sweden.[20]

We find that the greatest effect of the combined PrEP intervention follows from the decreased susceptibility to HIV, not the increased HIV testing rate. This result would be hidden in a model not taking the different stages of infection into account (see online supplementary material S5). Hence, to make a correct assessment of a PrEP programme's effect, the complexities of HIV transmission, the different stages of infection, need to be accounted for.

The benefit of targeting high-risk individuals for PrEP has been found by other studies.[8 21–24] Our analyses add to these findings by including additional parameters. Punyacharoensin et al[8] investigate the effect of different HIV interventions, including PrEP, among MSM in the UK. They define low actives as MSM with one or fewer new sexual partners a year, while our definition of high active MSM (at least 15 partners a year) is to address a group with very high HIV risk. Second, they address mixing through a different method using an OR among male heterosexuals, while we use data from the MSM population under study. Rozhnova et al[24] use four risk groups, however, they do not estimate mixing between the groups but assume intermediate mixing.

Our model has four strengths worth mentioning. First, our model design is strengthened by that it is calibrated to fit detailed data of MSM who visited an STI/HIV testing clinic in Sweden. For example, when the model is calibrated to data and the observed 5% HIV prevalence, the estimated percentage of individuals with HIV that are on ART treatment, 95.8%, is very close to the observed value of 95.1%.[1] In addition, the very same clinic where the data was gathered is the largest implementer of PrEP in Sweden, prescribing PrEP to 315 MSM since October 2018. Second, we can measure the assortativity with respect to activity degree of the study participants. The mixing between high actives and low actives is estimated to be more assortative than proportionate mixing (0.14 vs 0), and by using the estimated assortativity we get more reliable results with narrower 95% CRIs than for proportionate mixing (see online supplementary material S5). For realistic mean times to ART, the mixing assumption has an impact on the estimated prevalence, making it an important factor to include. Third, an important model choice is to include steady partnerships, not only casual contacts, since HIV transmission occurs to a large extent

within a steady partnership (table 2). Finally, the result concerning PrEP coverage is robust to variations in the parameter set-ups (see online supplementary material S5).

Our model includes limitations. First, the proxy question used to fit the assortativity can only define two activity degree groups and not more. The real-life scenario is probably more heterogeneous than accounted for in our model; even with a high PrEP coverage, the prevalence would likely stay above 0% due to some subgroups of MSM taking larger HIV risks than the high actives within our model. Additionally, some individuals are probably even more low active (such as MSM in monogamous steady partnerships) than we allow for. Second, many possible changes in sexual behaviour are not included. Our model assumes no change in sexual behaviour when being on ART and we do not assume any increased HIV testing rate for a sex partner to someone living with HIV. Individuals on PrEP are assumed to stay on PrEP, except if they get diagnosed with HIV and are put on ART treatment. Moreover, individuals are assumed to belong to one of the sexual activity groups during their whole life. For the studied MSM population we do not have data on how individuals change between the two activity groups, due to that the timeline data were restricted to 12 months. To study the potential effect of behavioural changes of activity degree we allowed low active MSM to become high active and for high active MSM to become low active. We looked at several different magnitudes of switching between low active and high active and these analyses can be found in online supplementary material S5. In this extension of the model we also allowed for PrEP users to stop taking PrEP. We find the same tendency as Rozhnova et al[25]—that a slightly less PrEP coverage is needed to reach a long-term HIV prevalence of 0% when allowing individuals to change activity degree group. When individuals are fixed to one group the PrEP coverage needed to eliminate HIV was 3.5% while when allowing for switching the PrEP coverage needed varies between 2.5% and 3%. Third, the data are collected among a convenience sample of MSM visiting an HIV/STI testing clinic, thereby it is not representative of all MSM in Sweden. Fourth, the model does not consider imperfect PrEP adherence which could overestimate PrEP's potential effect in reducing the HIV prevalence. However, in online supplementary material S5 we consider different values of PrEP effectiveness. Finally, our model does not incorporate concurrent steady partnerships. This is a common assumption for compartmental models,[8 15 26] and inclusion would possibly strengthen the model. However, our model does consider casual sex partners concurrent to steady partners.

In future work, risk compensation could be studied more thoroughly, for example, changed behaviour of individuals on PrEP. Other risk behaviours for HIV than sexual activity degree could be considered to define the risk group offered PrEP, such as taking part in group sex, consistent drug use and transactional sex. Another possible extension is to stratify our model by age, letting also activity degree vary between the age groups to capture that certain age groups could be more sexually active.

We conclude by stating the result emerged from the heterogeneous activity degrees: heterogeneity in sexual activity does increase the prevalence, however, it also makes targeted interventions much more effective.

**Acknowledgements** The authors thank the study participants for their contribution and acknowledge the work of staff at Venhälsan STI/HIV clinic in Stockholm, Sweden.

**Contributors** DH, SS, KYL and TB conceived the study. SS designed and managed the gathering of data. DH, TB and KYL defined the model. DH, SS and TB drafted the manuscript. All authors approved the manuscript before submission for publication.

**Funding** The data collection was financially supported by the Swedish State Grant for HIV/STI prevention (HSN 1309-1016) and an unrestricted grant from Gilead Sciences Nordic Fellowship Programme 2014. TB and KYL are supported by the Swedish Research Council (grant number 2015-05015).

**Disclaimer** The funders had no role in study design, data collection, analysis or write-up of results.

**Competing interests** None declared.

**Patient consent for publication** Not required.

**Ethics approval** Ethical approval for the study data collection was obtained from the Regional Ethical Committee in Stockholm, Sweden (Dnr 2014/1729-31/5).

**Provenance and peer review** Not commissioned; externally peer reviewed.

**Data availability statement** Data are available upon reasonable request. The full data set will be shared upon reasonable request to SS in order to protect participant confidentiality. This is motivated by that the data set contains sensitive information on sexual behaviour on a rather small sample of a stigmatised population (MSM), and sociodemographic data might theoretically make the data traceable.

**ORCID iD**
Disa Hansson http://orcid.org/0000-0003-1701-9325

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
