## [Reviewer comments · BMJ Open]

ARTICLE DETAILS

TITLE (PROVISIONAL)	Introducing pre-exposure prophylaxis to prevent HIV acquisition among men who have sex with men in Sweden: insights from a mathematical pair-formation model
AUTHORS	Hansson, Disa; Strömdahl, Susanne; Leung, Ka Yin; Britton, Tom

VERSION 1 – REVIEW

REVIEWER	Ian Spicknall Centers for Disease Control and Prevention, Division of STD Prevention. USA
REVIEW RETURNED	09-Oct-2019

GENERAL COMMENTS	This is my second time reviewing this manuscript. The authors have addressed most of my previous concerns. I appreciate how the authors have explored how different mixing patterns, combined with different PrEP targeting leads to different intervention effects. This is all well thought out, as I previously said. With that said, I believe there is still one large limitation (that the authors acknowledge in their limitations section) that could challenge the primary inferences of this paper: that people's behavioral category (high versus low active) is static. This could be argued to be true under short term scenarios, but the authors are performing equilibrium analyses over vast time periods. To strengthen this, the authors could take either of the following paths: 1) choose to focus on a shorter time frame, during which static behavior might make more sense, or 2) the authors could incorporate some behavioral volatility, where people change from low to high and vice versa at specified rates, likely holding constant the overall proportion of high active people and still conduct their equilibrium analysis. Did the authors see individual level behavior change from the time series of behavioral data? This could be used as rationale for ignoring behavioral dynamics over a 1 year period, but it still does not help with ignoring behavioral dynamics while estimating equilibrium infection dynamics. Another issue I raised previously has do with examining short-term dynamics. I appreciate that the authors present the short-term results in the supplementary materials. Are qualitative associations similar in the short term to that described in the long term? This could easily be noted in the manuscript text. It would also be helpful if the authors characterized roughly the amount of time required to achieve a new (pseudo) equilibrium. Currently, it is only specified as a "long time". Is this decades, centuries, millennia? I appreciate the limitations given in the discussion section. However, more discussion is needed for the potential effect of each limitation. This is especially true for the lack of PrEP
--

	dynamics and dynamic sexual behavior; for example, targeting of high activity MSM will become less effective as the duration of time spent in the high activity state decreases; in other words the faster the turn-over, the greater departure from the results presented. In addition, the effect of ignoring PrEP dynamics (that people remain on PrEP once they initiate, unless they either die or acquire HIV) should also be discussed more, rather than only mentioned; how might ignoring PrEP dynamics affect the authors' qualitative inferences. Minor suggestion: In the results of the abstract, I suggest adding "HIV" to the phrase 'long-term (HIV) prevalence.' HIV is only mentioned once before in the abstract (in the first sentence), and it is possible that PrEP may have indirect effects on the transmission dynamics of other sexually transmitted infections.
--	---

VERSION 1 – AUTHOR RESPONSE

Response to comments given by the reviewer

This is my second time reviewing this manuscript. The authors have addressed most of my previous concerns. I appreciate how the authors have explored how different mixing patterns, combined with different PrEP targeting leads to different intervention effects. This is all well thought out, as I previously said.

With that said, I believe there is still one large limitation (that the authors acknowledge in their limitations section) that could challenge the primary inferences of this paper: that people's behavioral category (high versus low active) is static. This could be argued to be true under short term scenarios, but the authors are performing equilibrium analyses over vast time periods. To strengthen this, the authors could take either of the following paths: 1) choose to focus on a shorter time frame, during which static behavior might make more sense, or 2) the authors could incorporate some behavioral volatility, where people change from low to high and vice versa at specified rates, likely holding constant the overall proportion of high active people and still conduct their equilibrium analysis. Did the authors see individual level behavior change from the time series of behavioral data? This could be used as rationale for ignoring behavioral dynamics over a 1 year period, but it still does not help with ignoring behavioral dynamics while estimating equilibrium infection dynamics.

*We chose to take path 2) suggested—to incorporate behavioural changes in the activity-degree but keeping the proportion high-active in the population constant. When we incorporated switching between activity-groups we additionally decided to include in the extension that a high-active on PrEP who switch to low-active cease to take PrEP. We chose this path since it seems more realistic and because we then could address two limitations raised by the reviewer: the lack of possible changes in sexual behaviours and the assumption that individuals on PrEP stays on PrEP their entire sexually active life. By doing this analysis, we found that a slightly less PrEP coverage is needed to obtain an equilibrium prevalence of 0%. This is the same tendency in results as in the paper by Rozhnova et al. (Impact of sexual trajectories of men who have sex with men on the reduction in HIV transmission by pre-exposure prophylaxis, *Epidemics*, 2019). They do however not consider a pair-formation model and do not estimate mixing between the different risk-groups. In the discussion we have added a paragraph on this (page 18, lines 7-17), but we also created a new section in the supplemental material section S4.8.*

Another issue I raised previously has do with examining short-term dynamics. I appreciate that the authors present the short-term results in the supplementary materials. Are qualitative associations similar in the short term to that described in the long term? This could easily be noted in the manuscript text. It would also be helpful if the authors characterized roughly the amount of time required to achieve a new (pseudo) equilibrium. Currently, it is only specified as a “long time”. Is this decades, centuries, millennia?

We have now added a paragraph to the previously sole sentence on this short-term result in the main text, page 15 lines 16-21. With the lowest possible PrEP-initiation rate that yields an equilibrium prevalence of 0% it would take 500 years before reaching no new HIV infections. We added this information to the supplementary material covering the short-term effect

I appreciate the limitations given in the discussion section. However, more discussion is needed for the potential effect of each limitation. This is especially true for the lack of PrEP dynamics and dynamic sexual behavior; for example, targeting of high activity MSM will become less effective as the duration of time spent in the high activity state decreases; in other words the faster the turn-over, the greater departure from the results presented. In addition, the effect of ignoring PrEP dynamics (that people remain on PrEP once they initiate, unless they either die or acquire HIV) should also be discussed more, rather than only mentioned; how might ignoring PrEP dynamics affect the authors' qualitative inferences.

In the additional analysis conducted we allowed for MSM on PrEP to become low-active and thereby to cease taking PrEP. We hope this is enough as a discussion on its potential effect.

Minor suggestion:

In the results of the abstract, I suggest adding “HIV” to the phrase ‘long-term (HIV) prevalence.’ HIV is only mentioned once before in the abstract (in the first sentence), and it is possible that PrEP may have indirect effects on the transmission dynamics of other sexually transmitted infections.

Thank you for noticing this. We have added the suggested “HIV” on page 2 line 20.